# Dendrimers as Nanocarriers for the Delivery of Drugs Obtained from Natural Products

**DOI:** 10.3390/polym15102292

**Published:** 2023-05-12

**Authors:** Huan An, Xuehui Deng, Fang Wang, Pingcui Xu, Nani Wang

**Affiliations:** 1Department of TCM Literature, Zhejiang Academy of Traditional Chinese Medicine, Hangzhou 310007, China; annhuan123@163.com (H.A.); xpc123ok@163.com (P.X.); 2School of Pharmacy, Zhejiang Chinese Medical University, Hangzhou 310007, China; dengxh1031@163.com (X.D.); w_ffang@163.com (F.W.)

**Keywords:** dendrimer, natural products, drug delivery, alkaloids, polyphenols

## Abstract

Natural products have proven their value as drugs that can be therapeutically beneficial in the treatment of various diseases. However, most natural products have low solubility and poor bioavailability, which pose significant challenges. To solve these issues, several drug nanocarriers have been developed. Among these methods, dendrimers have emerged as vectors for natural products due to their superior advantages, such as a controlled molecular structure, narrow polydispersity index, and the availability of multiple functional groups. This review summarizes current knowledge on the structures of dendrimer-based nanocarriers for natural compounds, with a particular focus on applications in alkaloids and polyphenols. Additionally, it highlights the challenges and perspectives for future development in clinical therapy.

## 1. Introduction

Natural products possess diverse pharmacological activities and low toxicity [1]. However, most drugs have poor bioavailability due to their hydrophobicity, making them unable to develop into ideal dosage forms. The development of dendrimer molecules can largely solve this. Dendrimers are three-dimensional, hyperbranched, monodispersed polymers composed of a central core, branches, and terminal functional groups attached to the branches [2,3]. They are considered ideal carriers for drug delivery due to their rich internal cavities, functionalized surfaces that introduce various chemical groups to achieve high customization, and good biocompatibility [4,5,6]. Compared with traditional polymers, dendrimers have the advantages of high water solubility, polyvalency, biocompatibility, and a precise molecular weight [7]. Various dendrimers have been developed and applied as drug delivery vehicles for natural products (Figure 1) such as polyamidoamine (PAMAM), polylysine (PLL), polypropylene (PPI), and polyglycerol (PG) [8]. The above dendrimers can be used for targeting drug delivery through various methods, such as intravenous, subcutaneous, intraperitoneal injection, oral, and ocular delivery [9].

Drugs are transported by dendrimers in two ways: (1) non-covalent interactions—dendrimers wrap drug molecules within the interior of dendrimers and protect them from being metabolized by the body when they reach their target location, thereby increasing their bioavailability; (2) covalent interactions—drugs are covalently linked to dendritic polymers, and the covalent bonds typically select cleavable functional groups such as esters, amines, and carbamates to effectively control drug release [10,11,12]. This review introduces the current status of dendrimer delivery systems for natural products, especially alkaloids and polyphenolic compounds (Table 1), and provides challenges and perspectives for the clinical development of the above dendrimers in the future.

## 2. Alkaloids Compound

Alkaloids are nitrogen-containing organic compounds which are divided into quinoline alkaloids, quinolizidine alkaloids, indole alkaloids, etc. [13]. These molecules have a wide range of biological activities, including antitumor, antibacterial, antiviral activities, etc. [14,15,16,17]. However, there are characteristics such as low solubility, instability, and low bioavailability in vivo. Typical bioactive alkaloids include camptothecin, paclitaxel, and berberine.

### 2.1. Camptothecin

Camptothecin (CPT) is an alkaloid isolated from *Camptotheca acuminata* which exhibits effective antitumor activity by targeting intracellular topoisomerase I enzyme and which is used to treat different types of cancer [18]. However, the bioavailability of CPT is unsatisfactory. The area under curve (AUC) and half-life of CPT (1 mg/kg, intravenous injection) were 2 × 10^−4^ mg h/mL and 1.291 h, respectively [19]. Additionally, the low water solubility, poor stability, and certain toxicity to normal cells limit the clinical use of CPT [20].

CPT can be encapsulated in the dendriform interior of PAMAM through hydrophobic interaction and can also be chemically combined with PAMAM dendrimers to achieve sustained drug release (Figure 2). Alibolandi et al. prepared a CPT-loaded PEGylated PAMAM G5 dendrimer and functionalized the carrier with AS1411 antinucleolin aptamers (1) to enhance the specific targeting to tumor cells and improve endocytosis [21]. N-acetyl-D-glucosamine has also been applied to increase the targeting of CPT-loaded PAMAM [22]. 

CPT has also been covalently conjugated to the surface of PAMAM dendrimers. CPT-conjugated PAMAM G4 dendrimers (2) can inhibit the growth of colorectal cancer cell line HCT-116 and induced nuclear fission [23]. Ma et al. selected a glucose transporter 1 (GLUT1)-specific ligand and glutathione (GSH)-sensitive junction to prepare a glucose-polyethylene glycol(PEG)-PAMAM-S-CPT-Cy7 conjugate to deliver CPT to GLUT1 overexpressed HepG2 liver tumor cells (3) [24]. CPT could be covalently linked to PAMAM G3 dendrimers by an acrylate end group. The conjugates are linked by PEG (4). CPT can be cleaved from dendrimers through the ammonolysis of ester bonds, and the rate of cleavage can be adjusted by pH. The drug delivery system prolongs the release time of the drug, and has injectability, which has a significant tumor inhibitory effect in head and neck cancer [25]. Furthermore, PLL dendrimers can be covalently linked to CPT. Fox et al. [26] prepared two CPT-bonded PLL dendrimers with glycines as the linker. In summary, PAMAM is the main carrier of CPT and has been used to develop several multifunctional nanomedicines.

### 2.2. Paclitaxel

Paclitaxel (PTX) is a kind of taxane diterpenoid compound used to treat breast cancer, ovarian cancer, pancreatic cancer, lymphatic cancer, etc. [27]. However, PTX is a hydrophobic substance and almost insoluble in water (3 × 10^−4^ mg/mL) [28]. PTX also has dose-dependent toxicity, including neurotoxicity, cardiovascular toxicity, gastrointestinal toxicity, and cutaneous toxicity [29].

In order to increase the specificity of PTX, the PTX-loaded PAMAM dendrimers are modified by ligands or active substances (Figure 3). The cationic PAMAM G3 dendrimers with dodecyl groups and diethylethanolamine (5) improved the effect of PTX on the inhibition of primary tumor growth and reduced tumor metastasis [30]. α-tocopheryl succinate (α-TOS) (6) can increase the targeting of PTX-PAMAM dendrimers [31]. Biotin (7), omega-3 fatty acid, alkali blue, and octa-arginine (R8) (8) can also be applied to increase the targeting of PTX-loaded PAMAM, subsequently enhancing the potential for cell uptake, cytotoxicity, and apoptosis [32,33,34,35,36].

Additionally, sialic acid (9), glucosamine (10), concanavalin A (11), and thiamine (12)-modified PPI dendrimers can significantly increase the transport amount of PTX, resulting in a higher biological distribution of PTX in brain tumors [37,38].

With regard to solubility enhancement, the triazine dendrimers had abundant hydroxyl groups (13) and increased solubility of PTX, from 0.0003 mg/mL to 0.562 mg/mL [39]. PG dendrimers enhanced the solubility of PTX by surrounding the aromatic ring of PTX and some methylene groups (14). The solubility increases with the increase in PG dendrimer generation [40].

The connection between drugs and carriers can affect the bioavailability of drug delivery platforms. Teow et al. selected a glutaric anhydride linker (15) to connect PTX to PAMAM G3 dendrimers; the permeability was 12 times that of PTX alone [41]. When the PTX was linked to PAMAM G4 dendrimers with succinic acid, the cytotoxicity of the conjugate was 10 times higher than that of PTX alone in A2780 human ovarian cancer cells [42]. Using the enzyme-sensitive linker glycylphenylalanylleucoglycine (GFLG) to connect PTX with dendrimers, the conjugate can specifically target cancer cells, causing significant cancer cell toxicity and apoptosis [43,44].

Small interfering RNA (siRNA) is widely used to silence malignant genes and has shown great prospects in cancer treatment [45]. In addition to their use in delivering siRNA, dendrimers have also been utilized to improve the delivery of PTX for the treatment of pancreatic and ovarian cancer. Studies have shown that these dendrimer–PTX complexes exhibit excellent loading capacity and targeting, leading to significant inhibition of tumor growth and cell apoptosis [46,47]. Overall, the use of dendrimers offers promising opportunities to enhance the targeting, transferability, solubility, and bioavailability of PTX for cancer therapy.

### 2.3. Berberine

Berberine (BBR) is a nitrogen-containing cyclic natural alkaloid [48,49]. Modern pharmacological studies have confirmed that BBR is used to treat various cancers, including breast cancer, lung cancer, liver cancer, ovarian cancer, cervical cancer, prostate cancer, etc. [50,51]. However, BBR has a very low absolute bioavailability (0.68%) [52]. The half-life and AUC of BBR in mice (0.5 mg/mL, intravenous injection) are 6.7 h and 1.424 mg/mL/h, respectively [53].

PAMAM dendrimers could increase the permeability of BBR and prevent its outflow from cancer cells [54]. The PAMAM dendrimer of BBR conjugation could increase the half-life and AUC by 2.1 and 1.7 times, respectively, and reduce the elimination rate constant [53]. BBR was delivered through PAMAM G4 dendrimer conjugation or encapsulation, and both formulations showed hemolytic toxicity of less than 5%, demonstrating safety and biocompatibility. Compared with encapsulation, conjugated PAMAM-BBR has stronger anticancer activity against MCF-7 and MDA-MB-468 breast cancer cells and a slower drug release rate. It is believed that using conjugation to deliver BBRs through dendrimers is better than using encapsulation [53]. 

## 3. Polyphenolic Compounds

Polyphenol compounds are widely found in natural plants, vegetables, and fruits. They mainly include phenolic acids, flavonoids, tannins, stilbene, and lignans [55,56]. Polyphenol compounds have antioxidant, anti-inflammatory, and antitumor activities as well as outstanding performance in the treatment of cancer, metabolic diseases, and other aspects. Their specificity, low toxicity, or non-toxicity are key advantages of dendrimers as anticancer agents, especially in the field of oncology [57,58,59,60,61]. However, their low bioavailability in humans limits their clinical applications [62,63,64].

### 3.1. Quercetin

Quercetin has antioxidant, anti-inflammatory, and antitumor pharmacological activities. The antitumor aspect involves breast cancer, liver cancer, colon cancer, etc. [65,66,67,68,69,70]. However, there are application limitations related to the stability of use in vivo, which limits the use of quercetin [71]. The aqueous solubility of quercetin is 0.000171 mg/mL [72].

PAMAM dendrimers can promote the solubility of quercetin. The main reason is that the amine groups of PAMAM dendrimers interact with quercetin. Another reason is that PAMAM dendrimers have sufficiently large internal cavities to capture the guest molecules of quercetin, and hydrophobic molecules dissolve in the aqueous medium. Pharmacodynamic studies showed that quercetin PAMAM G3 dendrimer (16) increased the anti-inflammatory activity of quercetin and extended the biological half-life of quercetin [73]. 

Targeted drug delivery can improve the therapeutic effect of cancer. Margetuximab can attach to the extracellular domain of the HER2 receptor on the surface of breast cancer, promote the effective internalization of nanocarriers, and be used as a targeting agent in drug delivery systems [74]. Yasaman et al. synthesized margetuximab and PEG conjugated PAMAM G4 dendrimer to deliver quercetin to MDA-MB-231 breast cancer cells (17). It has an obvious inhibitory effect on breast cancer cells by enhancing the expression of apoptosis genes bax and caspase 9 [75]. Seyed et al. composed hyperbranched PAMAM-PEG-folic acid-modified Fe_3_O_4_-nanoparticle-loaded quercetin (18). Due to the intracellular endocytosis mediated by the folic acid receptor, nanoparticles have significant targeting, selectively enter cancer cells to release drugs, and improve the anticancer effect [76]. Figure 4 describes typical quercetin-loaded dendrimers.

### 3.2. Gallic Acid

Gallic acid (GA) is a kind of phenolic acid which shows obvious anticancer activity in various cancers, such as colon cancer, breast cancer, lung cancer, stomach cancer, liver cancer, etc. GA has been recognized as a potential anticancer agent [77,78].

PAMAM G4 dendrimers can increase the bioavailability of GA and its antitumor activity against colon cancer cells HCT116. Its mechanism is to increase the uptake of GA [79]. At the same time, dendrimers target the release of GA and play a synergistic role with anticancer drugs [80]. After oral administration of the conjugation to a CCl_4_-induced oxidative damage rat model, the conjugation significantly reduced liver marker enzymes and enhanced the protection of GA to affect the liver. It may be related to dendrimers controlling the release rate of GA, allowing GA to be continuously released and maintaining the minimum effective concentration of the drug for a longer period of time, thereby improving the bioavailability [81].

### 3.3. Resveratrol

Resveratrol (RSV) is a kind of stilbene compound and an obvious antioxidant. RSV’s pharmacological applications include anticancer, diabetes, vascular metabolic diseases, etc. [82,83,84]. Due to its low water solubility, short half-life (9.2 ± 0.6 h), and low bioavailability, the current use of RSV is insufficient [85,86,87].

Due to electrostatic interactions and hydrogen bonding, RSV can be encapsulated by dendritic macromolecules, which can solve the shortcomings of RSV. PAMAM G4 dendrimers can improve the solubility and stability of RSV in water and cream formulations and enhance its penetration in the skin [88]. Sugary maze dendrimer-like glucan (SMDG) increased the solubility of RSV; thus, the antioxidant activity and cell uptake ability of RSV were also significantly enhanced [89]. Octenylsuccinate hydroxypropyl phosphoglycogen (OHPP) dendrimer decreased the crystallinity of RSV and increased the solubility in a dose-dependent manner. This was mainly due to hydrophobicity and hydrogen bonding between OHPP and RSV. Specifically, the high-density distributed hydroxypropyl and octenylsuccinate groups on the surface of OHPP have strong interactions with RSV, providing a favorable environment for this compound [90].

### 3.4. Silybin

Silybin (SIL) is a natural flavonoid lignan isolated from the plant of Silybin. SIL has antioxidant and anti-inflammatory properties as well as antitumor effects [91]. Unfortunately, it has poor water solubility (0.4 mg/mL). The oral bioavailability of SIL in rats is approximately 0.73% [92,93].

PAMAM dendrimers can significantly improve the water solubility and bioavailability through the electrostatic interaction between the external amines and the phenolic hydroxyl groups of the SIL [94]. PEGylated PAMAM-G4 dendrimers could increase the solubility of insoluble drug SIL. The results showed that the PEGylated system with a 2.0 kDa chain increased the solubility of SIL by 5 times [95]. Shetty et al. found that peptide dendrimers can enhance the skin penetration and deposition of antioxidant SIL [96].

### 3.5. Curcumin

Curcumin (CUR) is a natural polyphenol compound derived from the rhizome of *Curcuma longa* [97,98,99]. CUR can treat various diseases including cancer, metabolic diseases, orthopedic diseases, cardiovascular diseases, etc. [100,101,102,103]. Several clinical experiments have proved that CUR is non-toxic and safe for the human body [104,105,106]. CUR (2 g/kg) administered orally to healthy humans produced a maximum serum concentration (C_max_) of only 6 × 10^−6^ mg/mL within 1 h, an AUC_0−t_ of 4 × 10^−6^ mg/h/mL, and no serum concentration of CUR detected in the serum [107]. CUR has rapid metabolism, a short half-life, and poor bioavailability [108].

PAMAM dendrimers are considered a suitable carrier for encapsulating CUR, which protects CUR and has low toxicity through their surface-located amine groups, resulting in a significant inhibitory effect on cancer cells [109]. When the PAMAM dendrimers are loaded with CUR (Figure 5), the solubility of CUR is 415 times that of free Cur [110]. PAMAM G3 dendrimer-CUR (19) enhanced the toxicity of CUR on different types of cancer cells (MDA-MB-231 breast cancer cells, U-251 human malignant glioma, head and neck squamous cell carcinoma cells, etc.) [111]. PAMAM G5 dendrimers with acetyl terminal groups increased the solubility of CUR by 200 times and promoted the production of reactive oxygen species (ROS) in human lung adenocarcinoma A549 cells [112]. CUR was encapsulated by dendrimers (10% amine and 90% hydroxy) and can reduce the cell activity of three glioblastoma cell lines (mouse GL261, rat F98, and human U87). By decreasing the amount of amine on the surface of PAMAM dendrimers, the effective utilization rate of CUR can be improved [113]. Nosrati et al. modified PAMAM G5 dendrimers with Fe_3_O_4_ nanoparticles and loaded CUR on the surface of the nanocarrier (20). The inhibitory effect of this composite on MCF7 human breast cancer cells was stronger than that of free CUR [114].

Mitochondrial function is crucial for the occurrence and development of cancer cells [115]. Kianamiri et al. prepared a CUR mitochondrial delivery system using triphenylphosphonium (TPP)–PAMAM G4 dendrimer conjugation CUR (21) and found that CUR co-located with the mitochondria of cancer cells, inducing apoptosis of liver cancer cells without affecting normal cells. It can resist tumors by inducing ROS and lipid peroxidation, thereby activating the signaling pathway of apoptosis [116]. CUR–PAMAM G4 dendrimers grafted Bcl-2 siRNA onto amine groups, These nanoparticles (22) were delivered to human cervical cancer (HeLa) cells and increased the solubility and stability of CUR [117].

In addition, CUR can improve osteoporosis by acting on multiple steps of osteoclast activation and differentiation [118,119]. Yang studied the use of hexachlorocyclotriphosphazene (HCCP) as a linker to couple CUR and PAMAM to make nanoparticles, and the loading capacity of CUR could reach 27.2%. This nanoparticle effectively inhibited the differentiation of osteoclasts in a dose-dependent manner and promoted osteogenesis [120].

### 3.6. Puerarin

Puerarin (PUE) is an isoflavone compound extracted from *pueraria lobata* which has been widely used in the treatment of eye diseases, but there is a problem of low bioavailability. PAMAM dendrimers could interact with PUE through a weak hydrogen bond. The slow release of PUE from the complex results in prolonged ocular retention time, thereby increasing the bioavailability of PUE [121], in Table 1. Liu et al. prepared PUE liposomes coated with PAMAM dendrimers, and complex can improve the corneal permeability of rabbit eyes [122]. The pharmacokinetic parameters C_max_, AUC, and elimination half-life of PUE-PAMAM G3 dendrimer complex after ocular infusion were 1.3, 2, and 2.7 times higher than those of the PUE solution, respectively [123]. Dendrimers can also improve the oral bioavailability of PUE. The main reason is that the electrostatic interaction between the amine groups on the surface of the PAMAM dendrimers and the phenolic hydroxy groups on PUE increases the solubility of PUE. Additionally, higher generations of PAMAM dendrimers are more likely to interact with the cornea than lower generations [124,125], and the solubility of the whole generation of PAMAM dendrimers (G2/G3) is much stronger than that of the half-generation (G1.5/G2.5) [126].

## 4. Conclusions

The most ideal characteristics for the development of natural product pharmaceuticals are high stability, good bioavailability, and specific targeting. Dendrimers can be used in promising strategies due to the size and structure of nanospheres, high water solubility, and multivalent surface properties, which can effectively deliver drugs through encapsu-lation or covalent methods. 

This review found that the most commonly used dendrimers for natural products are the relatively mature PAMAM dendrimers, followed by PPI dendrimers, PPL dendrimers, PG dendrimers, and triazine dendrimers, which are used to treat various cancers (breast cancer, glioma, lung cancer, cervical cancer, ovarian cancer, pancreatic cancer, etc.), eye diseases, and osteoporosis. The above dendritic macromolecules have active groups such as amino and hydroxyl groups, which are suitable for drug covalent binding and physical adsorption and also provide rich modification sites for targeted therapy. The linkers between drugs and carriers can also affect the bioavailability of drug delivery platforms, such as GFLG, modified Fe_3_O_4_, succinic acid-glycine, and HCCP linkers, to achieve a pH response and enzyme-triggered transcytosis. In order to increase the specific targeting of natural product delivery platforms, researchers have developed various conjugated ligands and active substances to co-deliver drugs such as siRNA, N-acetyl-D-glucosamine, α-TOS, folic acid, Biotin, AS1411 antinucleolin aptamers, alkali blue, sialic acid, glucosa-mine, concanavalin A, thiamine, R8, transferrin, etc. Thus, dendrimers have been used as a platform for natural products and multiple ligands. However, the dendrimers currently used in natural products have a single structure, often only amino groups or hydroxyl groups, and there is no mixed-carrier matrix with multiple functional groups. Although these single-structured dendrimers are currently used for the delivery of some natural products, we need to specifically design and synthesize dendrimers with amino, carboxyl, and hydroxyl groups based on different drug solubility and targeting requirements. For example, the solubility and cell internalization of PEGylated dendrimers is increased by the change of the ratio between surface amino and hydroxyl groups. Thus, these multifunctional drug delivery systems will be able to better utilize the pharmacological effects of natural products.

Another limitation of dendrimers in the application of natural products is that there are currently only a few natural compounds used. This may be one reason why dendritic macromolecule-modified natural product complexes have not yet been used in clinical practice. Natural products have low toxicity and good safety, and have unique advantages in many chronic diseases, such as diabetes, obesity, hypertension, digestive diseases, etc. The natural products modified by dendritic macromolecules may provide better therapeutic effects in the treatment of these diseases. In addition, current research on natural products loaded with dendrimers mainly focuses on their pharmacological effects, and further exploration of their pharmacological mechanisms is needed.

In summary, dendrimers can effectively overcome the drawbacks of low water solubility, low bioavailability, and poor targeting of natural products. In order to develop more perfect drug delivery dendrimers for natural products in the future, it is necessary to design and synthesize multifunctional surfaces of dendrimers, select suitable ligands, and couple specific targeted substances to construct a drug delivery system to meet the needs of natural products with different structures and the targeted treatments of different diseases.

## Figures and Tables

**Figure 1 polymers-15-02292-f001:**
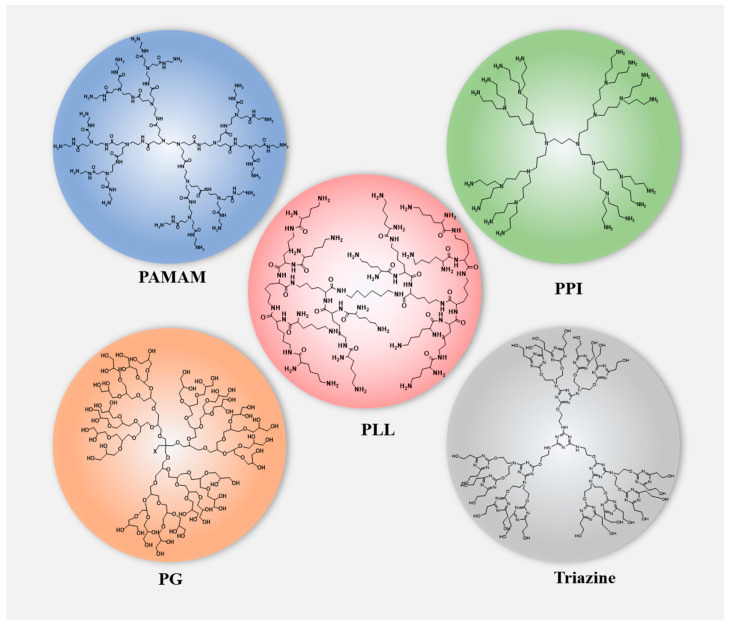
Structures of dendrimers that are used in drug delivery for natural products.

**Figure 2 polymers-15-02292-f002:**
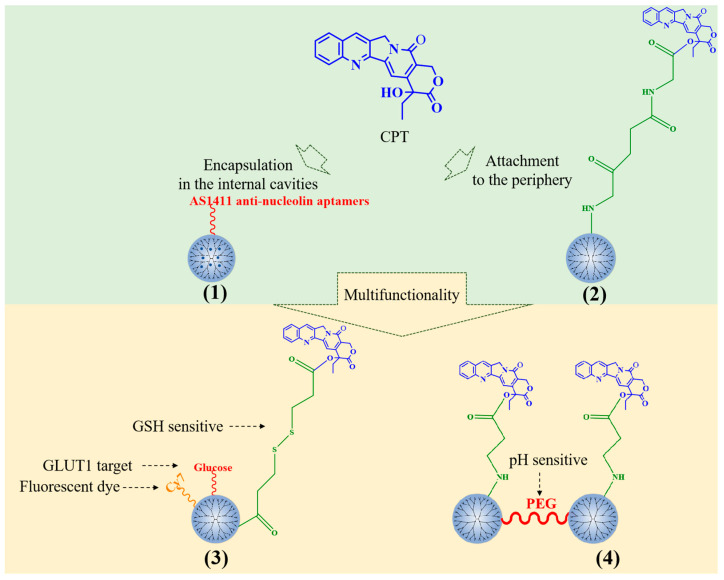
Typical CPT-loaded PAMAM dendrimers.

**Figure 3 polymers-15-02292-f003:**
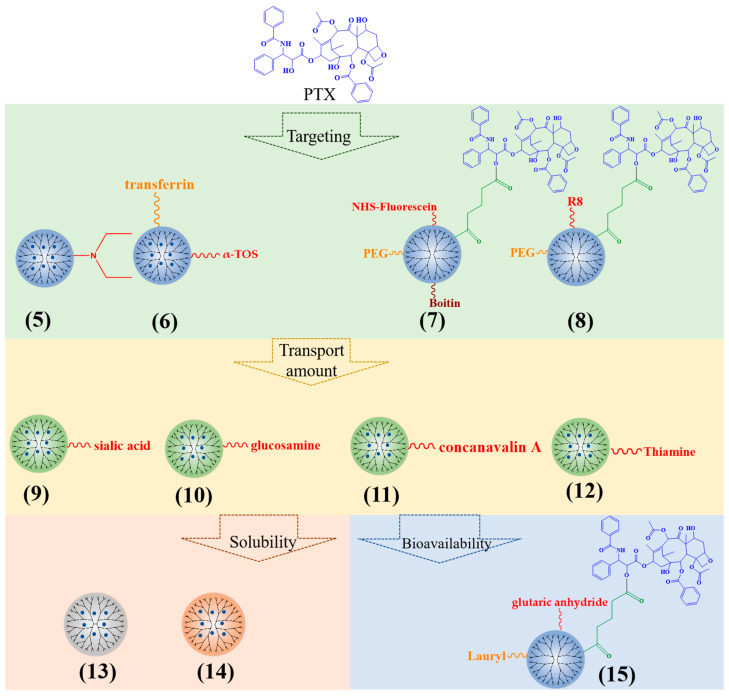
Typical PTX-loaded dendrimers.

**Figure 4 polymers-15-02292-f004:**
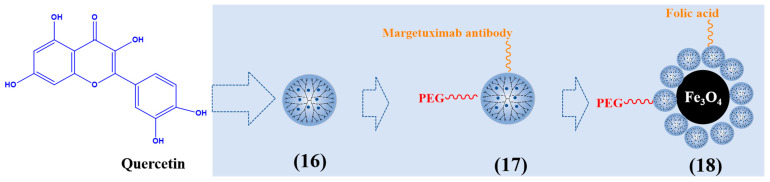
Typical quercetin-loaded dendrimers.

**Figure 5 polymers-15-02292-f005:**
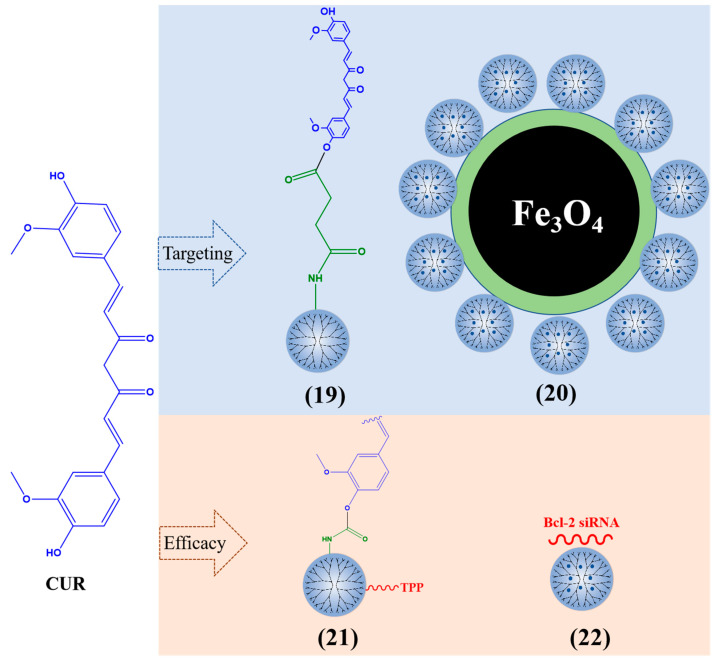
Typical CUR-loaded dendrimers.

**Table 1 polymers-15-02292-t001:** Characteristics of dendrimers for different applications.

No.	Type of Compounds	Compounds	Type of Dendrimers	Characteristic	Application	Mode of Administration	Refs.
1	alkaloid	CPT	PEGylated PAMAM G5 dendrimers–AS1411 antinucleolin aptamers (Encapsulation efficiency (EE) = 93.67%, loading content% (LC) = 8.1%)	specific targeting	BALB/c female mice bearing C26 tumors	intravenous injection (200 mL)	[21]
2	alkaloid	CPT	N-acetyl-D-glucosamine-labelled–PAMAM G3.5 dendrimers (EE = 66.26 ± 2.72%)	specific targeting	mice with B16F10 lung metastasis	intravenous injection	[22]
3	alkaloid	CPT	amine-terminated PAMAM G4 dendrimers–succinic acid-glycine linker	absence of small molecular weight impurities, size and drug content	colorectal cancer cells HCT-116	in vitro experiments	[23]
4	alkaloid	CPT	Glucose–PEG–PAMAM–S–CPT–Cy7	specific targeting, cellular microenvironment responsive	monolayer (2D) and multilayer tumor spheroid (3D) HepG2 cancer cell models GLUT	in vitro experiments	[24]
5	alkaloid	CPT	PAMAM G3 dendrimers–CPT with acrylate end groups	self-cleaving mechanism	HN12 head and neck tumor-bearing mouse	injection (0.16 mg of CPT in 50 μL/mouse)	[25]
6	alkaloid	CPT	PEGylated PLL dendrimers–glycine linker	improved the bioavailability, solubility, and efficacy of CPT	mouse (C26) and human colon cancer cells HT-29	intravenous injection (single dose of CPT 10 mg/kg)	[26]
7	alkaloid	CPT	PAMAM dendrimers–ROS–cleavable thioketal linker–CPT–surface modification with GGT	enzyme-triggered transcytosis	mice inoculated with BxPC-3 orthotopic pancreatic ductal adenocarcinoma	intravenous injection (equivalent to CPT 10 mg/kg)	[127]
8	alkaloid	PTX	dodecyl groups and diethylethanolamine surface-modified cationic PAMAM dendrimers	its high DNA binding ability and TLR inhibition activity, low toxicity, and smaller nanoparticle (NP) size.	a murine breast cancer metastasis model	intraperitoneal injection (15 mg/kg)	[30]
9	alkaloid	PTX	PAMAM G4 dendrimers–α-TOS–PEGylated Transferrin (EE = 71.18 ± 2.38%)	specific targeting	human cervical epithelial cells HeLa	in vitro experiments	[31]
10	alkaloid	PTX	PEGylated PAMAM G4 dendrimers–α-TOS	specific targeting	murine melanoma cancer cells (B16F10) xenografted C57Bl6/J mice	intratumoral injections (10 mg/kg/day)	[128]
11	alkaloid	PTX	PEGylated PAMAM G4 dendrimers–Biotin	specific targeting	human non-small cell lung cancer A549 cell line	in vitro experiments	[32]
12	alkaloid	PTX	PAMAMG4.0-NH2 dendrimers–omega-3 fatty acid docosahexaenoic acid	specific targeting	upper gastrointestinal cancers cells AGS and FLO-1	in vitro experiments	[33]
13	alkaloid	PTX	PAMAM-alkali blue dendrimers	the intralymphatic targeting	tumor-bearing mice	subcutaneous administration (1 mg/kg)	[34]
14	alkaloid	PTX	PEGylated PAMAM G4 dendrimer–PEG–R8	intracellular targetability	human cervical cancer cell line HeLa	in vitro experiments	[35]
15	alkaloid	PTX	PEGylated PAMAM G4 dendrimers–R8–vitamin-E succinate	cell penetration and improved PTX-mediated cytotoxicity	B16F10 tumor-bearing mice	intraperitoneal injection (10 mg/kg)	[36]
16	alkaloid	PTX	Sialic acid/glucosamine/concanavalin A/thiamine–PPI G5.0 dendrimers	specific targeting	human astrocytoma cells U373MG/human neuroblastoma cells IMR-32	in vitro experiments	[37,38]
17	alkaloid	PTX	triazine dendrimers	no adverse toxicity, increases its water solubility	BALB/c mice	intraperitoneal injection (40/60/100/200/500 mg/kg)	[39]
18	alkaloid	PTX	PG dendrimers	solubilization	different solvents	in vitro experiments	[40]
19	alkaloid	PTX	lauryl chains-modified PAMAM G3 dendrimers–glutaric anhydride linker	high permeability	human colon adenocarcinoma cell line Caco-2, primary cultured porcine brain endothelial cells	in vitro experiments	[41]
20	alkaloid	PTX	PAMAM G4 dendrimers–succinic acid linker	cytoplasmic and nuclear delivery, enhanced anticancer activity of PTX	human ovarian carcinoma cells A2780	in vitro experiments	[42]
21	alkaloid	PTX	Janus PEGylated peptide dendrimers–GFLG	enzyme-responsive feature	murine breast cancer cells 4T1 (tumor-bearing mice)	intravenous injection (5 mg/kg body weight each day for 10 days)	[43]
22	alkaloid	PTX	PAMAM G4 dendrimers–GFLG	specific targeting	breast cancer cell (MDA MB-231) mouse	intraperitoneal injection (40 mg/kg)	[44]
23	alkaloid	PTX	dendrimers–plectin-1 targeted peptide –nuclear receptor siRNA	tumor-targeted, redox-sensitive	panc-1 xenograft-bearing mice	intravenous injection	[46]
24	alkaloid	PTX	a triethanolamine-core PAMAM G6 dendrimers–Akt siRNA	initiating Akt target gene silencing both in vitro and in vivo, while being minimally toxic	mice containing human ovarian cancer cells SKOV-3	intraperitoneal injection (2 mg/kg/week)	[47]
25	alkaloid	PTX	phospholipid-modified PAMAM dendrimers–siMDR1	siRNA encapsulation ability, high gene delivery efficiency, and great cellular uptake	human breast cancer cells MCF-7/ADR	in vitro experiments	[129]
26	alkaloid	PTX	PAMAM G5 dendrimers–miR-21 inhibitor	improved the cytotoxicity of PTX increased the level of apoptosis of MCF-7 cells, decreased the invasiveness of the tumor cells	human breast adenocarcinoma cells MCF-7	in vitro experiments	[130]
27	alkaloid	BBR	PEGylated PAMAM G4 dendrimers (EE = 69.56 ± 23%)	controlled the release of drug, enhanced its bioavailability	human breast cancer cells MCF-7	in vitro experiments	[54]
28	alkaloid	sinomenine	hydroxy PAMAM G4 dendrimers with ethylenediamine nucleus (64 hydroxyl terminal groups)	increasing the therapeutic window in the treatment of early inflammation and for improving the efficacy of the drug in TBI	rabbit model of pediatric traumatic brain injury	intravenous injection (55 mg/kg, 200 μL)	[131]
29	polyphenol	quercetin	PAMAM G3 dendrimers	enhancing longer biological half-life	rats using a carrageenan-induced paw edema model	oral administration (20 mg/kg body weight)	[73]
30	polyphenol	quercetin	PEGylated PAMAM G4 dendrimer–Margetuximab	specific targeting	human breast cancer cells MDA-MB-231	in vitro experiments	[75]
31	polyphenol	quercetin	PAMAM-b-PEG-folic acid-modified Fe_3_O_4_ nanoparticles	pH-responsiveness, specific targeting	HeLa human cervical cancer cells, human breast cancer cells MDA-MB-231	in vitro experiments	[76]
32	polyphenol	GA	PAMAM G4 dendrimers	specific targeting, improved the bioavailability	human colon carcinoma cells HCT-116, human breast cancer MCF-7	in vitro experiments	[79,80]
33	polyphenol	GA	PAMAM-G4-NH2 dendrimers	improved the bioavailability, increased hepatoprotective effect	CCl_4_-induced oxidative damage in rat liver	oral administration (50 mg/kg/day, 7 days)	[81]
34	polyphenol	RSV	PAMAM G4 dendrimers	enhanced solubility, stability and transdermal permeation	simulated gastric and simulated intestinal fluid, rat skin	in vitro experiments	[88]
35	polyphenol	RSV	SMDG	improved bioavailability	human intestinal cells Caco-2	in vitro experiments	[89]
36	polyphenol	RSV	OHPP dendrimers	solubilization	solvents included ethanol, methanol, isopropanol, chloroform, acetonitrile, butanol, dimethyl sulfoxide (DMSO), pyridine, simulated gastric fluid, McIlvaine buffers	in vitro experiments	[90]
37	polyphenol	SIL	PAMAM G2 dendrimers	improved bioavailability	rats	oral administration (12 mg/kg)	[94]
38	polyphenol	SIL	PEGylated PAMAM G4 dendrimers	solubilization	DMSO	in vitro experiments	[95]
39	polyphenol	SIL	peptide dendrimers	enhanced skin permeation and deposition	rat skin	in vitro experiments	[96]
40	polyphenol	CUR	PAMAM G0.5 dendrimers/CUR (1:1/1:0.5)	improved solubility	mixture of distilled water and EtOH	in vitro experiments	[110]
41	polyphenol	CUR	PAMAM G3 dendrimers	full solubility, specific targeting, minimizing systemic toxic effect	breast cancer cells MDA-MB-231, human malignant glioma U-251, squamous head and neck cancer cells HNSCC, breast cancer cell line T47D	in vitro experiments	[109,111]
42	polyphenol	CUR	PAMAM G5 dendrimers with acetyl terminal groups	improved solubility and bioavailability	human lung adenocarcinoma cells A549	in vitro experiments	[112]
43	polyphenol	CUR	PAMAM dendrimers (10% amine and 90% hydroxyl-G4 90/10-Cys)	safe, only toxic to cancer cells	glioblastoma cell lines: mouse-GL261, rat-F98, and human-U87	in vitro experiments	[113]
44	polyphenol	CUR	PAMAM G5 dendrimers–modified citric acid coated Fe_3_O_4_ (EE = 45.58 ± 0.41%, LC = 12 ± 0.03%)	pH-responsiveness	human breast cancer cell line MCF7	in vitro experiments	[114]
45	polyphenol	CUR	PAMAM G4 dendrimers–TPP	mitochondrial targeting	HuH-7, Jurkat T cell, Hepa1-6, and human and mouse fibroblasts	in vitro experiments	[116]
46	polyphenol	CUR	PAMAM G4 dendrimers–Bcl-2 siRNA (LC = 82 wt%)	improved solubility and bioavailability, induced the most apoptosis in HeLa cancer cells	HeLa cells	in vitro experiments	[117]
47	polyphenol	CUR	PAMAM dendrimers–HCCP linker	pH-responsiveness	bone marrow macrophage cells BMMs	in vitro experiments	[120]
48	polyphenol	CUR	PAMAM G4 dendrimers–palmitic acid core–shell nanoparticle (EE = 80.87%)	potentially active against acute stress	adult male albino mice	intravenous injection (25 mg/kg)	[132]
49	polyphenol	PUE	PAMAM G3.5/G4/G4.5/G5 dendrimers	improved the corneal permeation	corneas of each male New Zealand albino rabbit	drip into the cornea (50 μL)	[121]
50	polyphenol	PUE	PAMAM G3/G4/G5 dendrimers	improved the corneal permeation	rabbit aqueous humor	instillation (0.5 mg, 1% PUE solutions)	[123,124,125]
51	polyphenol	PUE	PAMAM G2 dendrimers	improved solubility and bioavailability	rats	oral administration (130 mg/kg PUE)	[126]
52	polyphenol	baicalin	PAMAM dendrimers–folic acid	specific targeting	HeLa human epithelial carcinoma cell line, human lung carcinoma cell line A549	in vitro experiments	[133]
53	polyphenol	daidzein	PAMAM G3 dendrimers, PPI G4 dendrimers	improved solubility, prolonged the delivery, and maintained the antioxidant activity	human breast cancer MCF-7, human lung carcinoma cells A549	in vitro experiments	[134]
54	polyphenol	anthocyanin	PAMAM dendrimers–silica	inhibiting the proliferative effects of Neuro 2A cancer cells, non-toxicity to the cells	Neuro 2A cancer cells	in vitro experiments	[135]

## Data Availability

The data presented in this study are available on request from the corresponding author.

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
