# Peer review of "Dendrimers as Nanocarriers for the Delivery of Drugs Obtained from Natural Products"

_polymers, 2023, doi:10.3390/polym15102292_

Round 1
Reviewer 1 Report
The authors described the application of dendrimers as nanocarriers for the delivery of drugs of natural origin. The manuscript can be considered for further processing after addressing the following queries:
1). I suggest a slight modification of the title. The title of the review manuscript may be modified to “Dendrimers as nanocarriers for the delivery of drugs obtained from natural products”.
2). The second sentence of paragraph one of the introduction should be revisited. There is the repetition of the word “due to”.
3). Figure 1-5 included in the manuscript are not legible. Please increase the size of the pictures so that readers can focus on the main aspects of the article.
4). The manuscript requires extensive English language editing to improve the quality of the manuscript.
Extensive English language editing is needed to improve the quality of the manuscript.
Reviewer 2 Report
PFA

Please find attached file
Reviewer 3 Report
Dear Respectful Editor,
Thank you for giving me the opportunity to review this review article titled "Dendrimers as Nanocarriers for Drug Delivery of Natural Products." (polymers-2373482). The article presents an interesting and timely topic on the use of dendrimers as nanocarriers for natural products with low solubility and bioavailability. Overall, the article is well-written and organized, and provides a comprehensive review of the current state-of-the-art of dendrimer-based nanocarriers for natural products. The advantages of dendrimers, such as their precise molecular structure, low polydispersity index, and multiple functional groups, are well-discussed. Additionally, the article presents a clear and concise overview of the applications of dendrimers in clinical therapy and highlights the challenges and perspectives for future development in this field. However, there are some issues that need to be addressed in order to improve the clarity and quality of the article. Below are some specific comments and suggestions for improvement:
· I would like to suggest some small changes to the abstract section of the article to improve its clarity and accuracy.
For example: Natural products have proven their value as drugs that can be therapeutically beneficial in the treatment of various diseases. However, most natural products have low solubility and poor bioavailability, which pose significant challenges. To solve these issues, several drug nanocarriers have been developed. Among these methods, dendrimers have emerged as vectors for natural products due to their superior advantages, such as controlled molecular structure, narrow polydispersity index, and availability of multiple functional groups. This review summarizes current knowledge on the structures of dendrimer-based nanocarriers for natural compounds, with a particular focus on applications in alkaloids and polyphenols. Additionally, the review highlights the challenges and perspectives for future development in clinical therapy.
· I would recommend paying attention to explaining abbreviations where they are first used in the text.
*Various dendrimers have been developed and applied as drug delivery vehicles for natural products (Figure 1), such as PAMAM, PLL, PPI, and PG[8].
* Polyglycol dendrimers enhanced the solubility of PTX by surrounding the aromatic ring of paclitaxel and some methylene groups (14).
· Please revise the manuscript for any grammatical and spelling errors to ensure that the content is clear and concise.
For example: Please revise the manuscript for any grammatical and spelling errors to ensure that the content is clear and concise.
*PTX is a kind of taxane diterpenoid compound, which is one of the widely used natural anti-cancer drugs, and is used to treat breast cancer cancer, ovarian cancer, pancreatic cancer, lymphatic cancer, etc.[27]
*the bis functional carboxylic acid Succinic acid was selected as the linker between PTX and PAMAM G4 dendrimers, In A2780 human ovarian cancer cells, the cytotoxicity of the conjugate was 10 times higher than that of PTX alone[42].
· The introduction section needs to be more concise and focused. Please revise the specific parts of the text that need to be clarified or that have connectivity problems?There appears to be any significant connectivity problem between the two sentences. However, it might be helpful to add a transitional phrase or sentence to more clearly connect the ideas.
For example: Several studies used dendrimers to deliver siRNA and PTX to treat pancreatic cancer and ovarian cancer. Nanoparticles exhibit excellent loading capacity and targeting, significantly inhibiting tumor growth and inducing cell apoptosis[46,47]. In summary, dendrimers are used to carry PTX in order to improve its targeting, transferability, solubility, and bioavailability.
Recommend: In addition to their use in delivering siRNA, dendrimers have also been utilized to improve the delivery of PTX for the treatment of pancreatic and ovarian cancer. Studies have shown that these dendrimer-PTX complexes exhibit excellent loading capacity and targeting, leading to significant inhibition of tumor growth and cell apoptosis (46, 47). Overall, the use of dendrimers offers promising opportunities to enhance the targeting, transferability, solubility, and bioavailability of PTX for cancer therapy.
For example: Especially in the field of anticancer, its specificity, low toxicity, or non-toxicity make it a key advantage of an anticancer agent[57-61].
Recommend:Its specificity, low toxicity, or non-toxicity are key advantages of dendrimers as anticancer agents, especially in the field of oncology [57-61].
· There are symbols in different languages in Figure 3, which shows typical PTX loaded dendrimers.
· Please check the units used in the text.
Example: half- life and AUC of BBR in mice (500 mg, intravenous injection) are 6.7h and 1424.42 μg/ml/h, respectively [53].
· I noticed a potential error in the text regarding the use of "Reveratrol" or "Resveratrol". Could you please double-check and clarify which term is intended to be used in the manuscript?
· The section on the advantages of dendrimers needs to be more detailed and specific. For instance, the article mentions that dendrimers have "multiple functional groups", but it does not explain how these groups can be used to enhance drug delivery.
· The article needs to provide more examples and case studies of dendrimer-based nanocarriers for natural products. The current section on "applications in clinical therapy" is too general and does not provide enough details on specific drugs or diseases.
· The conclusion section needs to be more substantial and informative. It should summarize the main findings and contributions of the article and provide some insights and recommendations for future research.
Overall, I believe that with some revisions and improvements, this article can make a valuable contribution to the field of drug deliver of natural products using dendrimer-based nanocarriers.
Best regards,
Dear Respectful Editor,
Thank you for giving me the opportunity to review this review article titled "Dendrimers as Nanocarriers for Drug Delivery of Natural Products." (polymers-2373482). The article presents an interesting and timely topic on the use of dendrimers as nanocarriers for natural products with low solubility and bioavailability. Overall, the article is well-written and organized, and provides a comprehensive review of the current state-of-the-art of dendrimer-based nanocarriers for natural products. The advantages of dendrimers, such as their precise molecular structure, low polydispersity index, and multiple functional groups, are well-discussed. Additionally, the article presents a clear and concise overview of the applications of dendrimers in clinical therapy and highlights the challenges and perspectives for future development in this field. However, there are some issues that need to be addressed in order to improve the clarity and quality of the article. Below are some specific comments and suggestions for improvement:
· I would like to suggest some small changes to the abstract section of the article to improve its clarity and accuracy.
For example: Natural products have proven their value as drugs that can be therapeutically beneficial in the treatment of various diseases. However, most natural products have low solubility and poor bioavailability, which pose significant challenges. To solve these issues, several drug nanocarriers have been developed. Among these methods, dendrimers have emerged as vectors for natural products due to their superior advantages, such as controlled molecular structure, narrow polydispersity index, and availability of multiple functional groups. This review summarizes current knowledge on the structures of dendrimer-based nanocarriers for natural compounds, with a particular focus on applications in alkaloids and polyphenols. Additionally, the review highlights the challenges and perspectives for future development in clinical therapy.
· I would recommend paying attention to explaining abbreviations where they are first used in the text.
*Various dendrimers have been developed and applied as drug delivery vehicles for natural products (Figure 1), such as PAMAM, PLL, PPI, and PG[8].
* Polyglycol dendrimers enhanced the solubility of PTX by surrounding the aromatic ring of paclitaxel and some methylene groups (14).
· Please revise the manuscript for any grammatical and spelling errors to ensure that the content is clear and concise.
For example: Please revise the manuscript for any grammatical and spelling errors to ensure that the content is clear and concise.
*PTX is a kind of taxane diterpenoid compound, which is one of the widely used natural anti-cancer drugs, and is used to treat breast cancer cancer, ovarian cancer, pancreatic cancer, lymphatic cancer, etc.[27]
*the bis functional carboxylic acid Succinic acid was selected as the linker between PTX and PAMAM G4 dendrimers, In A2780 human ovarian cancer cells, the cytotoxicity of the conjugate was 10 times higher than that of PTX alone[42].
· The introduction section needs to be more concise and focused. Please revise the specific parts of the text that need to be clarified or that have connectivity problems?There appears to be any significant connectivity problem between the two sentences. However, it might be helpful to add a transitional phrase or sentence to more clearly connect the ideas.
For example: Several studies used dendrimers to deliver siRNA and PTX to treat pancreatic cancer and ovarian cancer. Nanoparticles exhibit excellent loading capacity and targeting, significantly inhibiting tumor growth and inducing cell apoptosis[46,47]. In summary, dendrimers are used to carry PTX in order to improve its targeting, transferability, solubility, and bioavailability.
Recommend: In addition to their use in delivering siRNA, dendrimers have also been utilized to improve the delivery of PTX for the treatment of pancreatic and ovarian cancer. Studies have shown that these dendrimer-PTX complexes exhibit excellent loading capacity and targeting, leading to significant inhibition of tumor growth and cell apoptosis (46, 47). Overall, the use of dendrimers offers promising opportunities to enhance the targeting, transferability, solubility, and bioavailability of PTX for cancer therapy.
For example: Especially in the field of anticancer, its specificity, low toxicity, or non-toxicity make it a key advantage of an anticancer agent[57-61].
Recommend:Its specificity, low toxicity, or non-toxicity are key advantages of dendrimers as anticancer agents, especially in the field of oncology [57-61].
· There are symbols in different languages in Figure 3, which shows typical PTX loaded dendrimers.
· Please check the units used in the text.
Example: half- life and AUC of BBR in mice (500 mg, intravenous injection) are 6.7h and 1424.42 μg/ml/h, respectively [53].
· I noticed a potential error in the text regarding the use of "Reveratrol" or "Resveratrol". Could you please double-check and clarify which term is intended to be used in the manuscript?
· The section on the advantages of dendrimers needs to be more detailed and specific. For instance, the article mentions that dendrimers have "multiple functional groups", but it does not explain how these groups can be used to enhance drug delivery.
· The article needs to provide more examples and case studies of dendrimer-based nanocarriers for natural products. The current section on "applications in clinical therapy" is too general and does not provide enough details on specific drugs or diseases.
· The conclusion section needs to be more substantial and informative. It should summarize the main findings and contributions of the article and provide some insights and recommendations for future research.
Overall, I believe that with some revisions and improvements, this article can make a valuable contribution to the field of drug deliver of natural products using dendrimer-based nanocarriers.
Best regards,
Round 2
Reviewer 2 Report
All the suggestions/corrections have been incorporated. Therefore I recommend this article for publication in Polymers.
Reviewer 3 Report
Dear Respectful Editor,
I recommend accepting this manuscript for publication in Polymers. The study's scientific quality, originality, and significance make it a valuable addition to the journal's content. I have no reservations or major concerns that would warrant any revisions or additional review rounds. I would like to commend the authors for their excellent work and congratulate them on their significant contributions to the field. I am confident that publishing this manuscript will benefit both the scientific community and the journal's readership.Thank you for considering my recommendation. Please do not hesitate to contact me if you require any further information or clarification regarding my review.
Yours sincerely,